# Gender gap in faculty promotion

Agata Czech[1☉], Marcelo Olarreaga[1¤☉]*, Olivia Peila[2☉]

1 Geneva School of Economics and Management, University of Geneva, Geneva, Switzerland, 2 Data and Statistics Office, University of Geneva, Geneva, Switzerland

¤ Current address: Geneva School of Economics and Management, University of Geneva, Geneva, Switzerland
☉ All these authors are contributed equally to this work.
* marcelo.olarreaga@unige.ch

**Data Availability Statement:** All data and is made available in OPEN ICPSR at https://www.openicpsr.org/openicpsr/project/209522/version/V2/view.

**Funding:** The author(s) received no specific funding for this work.

## Abstract

We examine the gender gap in faculty promotion at the University of Geneva. After building a new measure of research quality that has no gender bias (i.e. men and women have, on average, the same level of research quality after we control for disciplines), we find that conditional on research quality, discipline and place where the PhD was obtained, women are 11 percent less likely to get promoted. The gender gap is almost three times larger for promotion from assistant to associate professor, suggesting that the mechanism at play is stronger for junior faculty. The gender gap is explained by the fact that an equal increase in research quality leads to a smaller increase in women's probability of promotion.

## Introduction

We assess the gender gap in faculty promotion at the University of Geneva from 2004 to 2023. Using a parametric hazard survival model, we estimate the gender gap in promotion hazards. After controlling for research quality, discipline, and whether or not the PhD was obtained at the University of Geneva, we find that female faculty members are 11 percent less likely to get promoted. In a subsample of assistant professors, we find that women are 31 percent less likely to get promoted to the associate professor level than men. Importantly, when we allow for the impact of research quality to vary by gender, we find that the gender gap is fully explained by the fact that an equal increase in research quality leads to a smaller increase in women's probability of promotion.

There are at least three reasons why these results are important. First, gender discrimination in salary is difficult to observe in the public university system of the University of Geneva, where salaries are predetermined depending on rank (assistant, associate, full) and number of years in the position. However, as suggested by [1], a large share of gender discrimination takes the form of "equal pay for equal work, but unequal work". Gender differences in faculty promotion result in unequal work, leading to de facto large wage differences, as the median full professor wage is 30 percent higher than the median assistant professor wage. The potential for unequal work in academia is high. Although women constitute 47 percent of PhD graduates in Europe, their representation diminishes significantly as they climb the academic ladder, with only 21 percent serving as full professors (see [2]). Switzerland is no exception.

**Competing interests:** The authors have declared that no competing interests exist.

Note, however, that the differences in flows may be vanishing, as suggested by [3, 4], who provide evidence of positive discrimination in more recent years.

Second, while the University of Geneva has taken significant steps to address the gender gap in faculty hiring, including the introduction of short-list quotas and the presence of an equality delegate in every hiring commission, the efforts regarding faculty promotion have been more timid. The results in this paper suggest that efforts should also focus on promotions, particularly at the junior level.

Finally, the identification of a gender gap in the way research quality translates into promotion suggests that more objective measures of research quality that do not suffer from gender biases should be considered. More importantly, this gender unbiased research quality measure should also be unbiasedly assessed.

We face two main challenges. First, while there is publicly available data on the hiring and promotion of researchers at the University of Geneva as well as their publication and citation records, the publicly available data on retirement and other reasons to leave the University of Geneva is incomplete. This challenge was overcome using administrative data that the University of Geneva made available.

Second, controlling for the quality of research output seems essential when assessing promotion decisions. Usually, researchers use the number of citations or papers as a proxy for the quality of research output. There is, however, important literature suggesting that both measures may be biased (see [2, 5, 6]). To overcome this, we propose a new measure: the number of citations per paper. The basic idea is that if the bias in the number of citations and the number of papers is proportional and identical, using a ratio gets rid of the gender bias. Correcting for these biases in measures of research quality is important because the main mechanism explaining the gender gap in promotion is the gender difference in the impact of research quality on the probability of promotion.

We are not the first to work on the gender gap in faculty promotions. [5] conducted a comprehensive study on hiring, publications, citations, and promotions. Utilizing an extensive database spanning 7484 universities across 130 countries for six-time cross-sections and around half a million observations, the study reveals gender gaps in the four domains. In terms of faculty promotion they find that the likelihood of promotion is 27 percent lower for women at the beginning of the 20th century. The gender gap declines to 22 per cent by the middle of the 20th century and is as low as 6 per cent for their subsample of prestigious universities by the end of the 20th century.

There are obvious data and methodological differences between their paper and ours. We use administrative data for the University of Geneva and they have a sample that spans over many countries. We use a proportional hazard model and they use a linear estimator. To address the gender gap in publication and citations when measuring the quality of research output, we use a measure that aims at correcting those biases. Nevertheless, our estimates for the gender gap at the University of Geneva at the beginning of the 21st century of 11 per cent is consistent with their findings.

An important difference is that they only consider promotions to the full professor level. Our sample allows us to focus on promotions from assistant to associate professor and found the gender gap to be larger. This suggests that efforts to address the gender gap in academic promotions should focus on junior faculty. It also provides an indication that the mechanisms at play for this unexplained gender gap are stronger for junior faculty.

[7] explore whether a higher presence of female evaluators in committees affects the likelihood of female faculty promotion. They use a sample of 100,000 applications for associate and full professorships in Italy and Spain and show that an increased number of women in evaluation committees lowers the likelihood of women being promoted.

An important literature also examines mechanisms that can explain the gender gap in promotion. [2] investigates whether higher standards are imposed on female researchers, which may ultimately lead to lower promotion probabilities. Using a measure of writing clarity and applying it to 9117 article abstracts published between 1950 and 2015 in selected journals, the study finds that papers authored by women are 1-6 percent better written than equivalent papers by men. Results suggest that elevated standards may adversely affect women's productivity, as they may spend more time revising old research, leaving less time for new work, resulting in fewer publications and, therefore, less chance of promotion.

Despite potentially better writing, women's work may not receive due credit. [8] study explores whether gender influences credit and attribution for group work using observational data and two experiments. Analysing CVs of economists up for tenure between 1985 and 2014 at 35 US PhD-granting universities, they find that accounting for research quality and other factors, men and women who predominantly solo author their work have similar tenure rates. However, an additional co-authored paper correlates with a 7.4 percent increase in tenure probability for men, compared to a 4.7 percent increase for women. This gap lessens when women co-author with each other, indicating that credit attribution is linked to the gender mix of co-authors.

[6] supports the idea that gender bias exists in the attribution of research. Using bibliometric data to examine gender biases in citation patterns within economics, she builds an omission index measuring the likelihood of omitting female-authored papers from the literature. She finds that omitted papers are 15-20 percent more likely to be female-authored.

These papers suggest different channels that can explain the gender gap, but they also suggest that the number of publications or citations as a measure of research quality is a biased measure as women will have fewer publications and citations. In our empirical methodology, we try to address these biases when measuring the quality of research output. We also examine whether our unbiased measurement of research quality affects the likelihood of promotion of female and male researchers differently.

## Materials and methods

### Ethics statement

A written consent was obtained from the University of Geneva's Research Ethics Board: CUREG-20231219-453-2.

We use a survival analysis model [9] to analyze the time to promotion conditional on gender, discipline, research quality, and whether the individual has a Ph.D. from the University of Geneva.

We first use Kaplan-Meier plots to illustrate how the probability of promotion, or rather its absence, evolves with time [10]. The survival function (i.e., the percentage of individuals who have not been promoted) at time $t$ is given by the non-parametric product limit estimator:

$$S(t) = \Pi_{i:t_i < t}\left(1 - \frac{p_i}{n_i}\right) \tag{1}$$

where $p_i$ is the number of individuals who got promoted at the time $t_i$ and $n_i$ is the number of individuals who have not gotten promoted up to time $t_i$. To test the proportional hazard assumption, we used log-log survival plots. A log–log survival curve is simply a transformation of an estimated survival curve that results from taking the natural log of an estimated survival probability twice. The proportional hazard assumption cannot be rejected if the log-log-survival curves are parallel (do not cross each other).

We then estimate proportional hazard survival models of the probability of promotion as a function of gender and other control variables. We choose between the semiparametric Cox model, and the parametric Weibull hazard models using the Akaike Information Criteria (AIC) where $AIC = -2LogLikelihood + 2(c + p + 1)$, where $c$ is the number of covariates which is identical across different models, and $p$ is the number of parameters necessary to estimate each model (0 for Cox and exponential and 1 for Weibull regressions). The difference between the semiparametric Cox model and the parametric Weibull model is that the latter assumes that promotions follow a certain Weibull distribution, whereas in the case of the Cox model the distribution of promotions is unknown. The AIC should be interpreted as the distance between the estimated and the unknown true model. A model statistically dominates another when it is more than 2 AIC units lower than the other, i.e., the estimated model is closer to the unknown true model. We also test whether control variables should be included using the likelihood ratio statistic ($-2(L_N - L_F)$ where $L_N$ is the LogLikelihood of the model with no control variables and $L_F$ is the LogLikelihood of the full model. Under the null hypothesis that the control variables should not be included, the test statistic has a chi-squared distribution with degrees of freedom equal to the number of control variables (Kleinbaum and Klein, 2012).

## Data

Data on new faculty hires and promotions by school (that we take as a proxy for discipline), gender, as well as the University where they obtained their PhD were obtained from the University's web pages. See https://www.unige.ch/lejournal/trajectoires/ for data from 2020 onwards and https://www.unige.ch/lejournal/trajectoires/ for data from 2004 to 2019. When information on the place where they obtained their PhD was missing, we complemented it with searches of researchers' home pages and LinkedIn. Data on retirement was obtained also from the University of Geneva web pages. See https://www.unige.ch/lejournal/trajectoires/departs-retraite/, but unfortunately this data is only publicly available from 2020 onwards. Thus we complemented with administrative data from the University of Geneva to identify researchers' retirement date before 2020, or any other kind of departure from the University of Geneva during the period 2004-2023. We exclude from the sample full professors, as by definition, they cannot obtain a promotion.

Data on research quality was taken from Google Scholar through the Publish or Perish app. We measure research quality as the log of the number of citations per paper. To reduce the influence of outliers and avoid dropping observations for researchers with no citations per paper, which may be prevalent in some disciplines at early stages of a career, we use an inverse hyperbolic sine transformation: $q = ln(n/p + \sqrt{(n/p)^2 + 1})$, where $q$ is our measure of research quality, $n$ is the number of citations, and $p$ is the number of papers. Research quality is measured at the time of entry into UNIGE (or in 2004 if the researcher was already in the UNIGE in 2004) because the idea is to capture the time-invariant intrinsic value of research quality, as the estimator used does not allow for time-varying covariates. One could also use research quality measured at the time of promotion or exit from UNIGE. The problem is that the reasons for entry into UNIGE are homogeneous, whereas the reasons for exit vary. It goes from moving to a job outside academia to moving to a lower-ranked university or a better-ranked university, so we would be capturing research quality at very different stages of researchers' careers. Note that When we replace researcher's quality at entry with researchers' quality at exit, we still obtain a positive and statistically significant impact on promotions, but the point estimate is much smaller and it is less precisely estimated.

The reason for using the number of citations per paper rather than simply the number of citations or the number of publications is that there is evidence of gender biases in these two

measures leading to fewer papers being published by women (see [2, 5]), and fewer citations (see [5, 6]).

Assuming that these two biases lead to a proportional and identical measurement error in observed citations (*n*) and number of papers (*p*), then the bias disappears from the measure of research quality (*n/p*) when taking their ratio. There is some evidence that this may not be such a bad assumption. When running a regression of the log of citations on gender and school fixed effects, we find, as expected, that women are 55 percent less cited than men. When running a regression of the log of papers on gender and school fixed effects, we find, as expected, that women have 49 percent fewer papers. These gender differences are large and statistically significant. However, when running a regression of research quality on gender and school fixed effects, the coefficient on gender is small (-0.07) and statistically insignificant, suggesting that our measure of research quality does not seem to suffer from a gender gap.

Table 1 provides summary statistics by school (i.e., discipline) and for the University of Geneva as a whole. On average, 38 percent of faculty who were not hired as full professors obtained a promotion during the period 2004-2023. It is important to note that 38 percent is not the success rate of those applying for promotion, but the share of the total population of associate and assistant professors that get eventually promoted at the University of Geneva. Only 37 percent of professors in the sample are female and 36 percent obtained their PhD at the University of Geneva. The average number of citations per paper is 41.

There is some important heterogeneity in all these variables across schools. For example, the schools of social sciences, translation and sciences have the lowest promotion rates, all below 30 percent. The schools of Economics&Management, Literary Arts and Law have the highest promotion rates, all above 40 percent. Similarly, the share of female researchers among faculty that is not full professor is below 31 percent in Medicine and Sciences but above 65 percent in Law and Translation. Finally, research quality also exhibits some important heterogeneity across disciplines, with the average number of citations per paper being below 10 in Law and Theology and above 49 in Sciences and Medecine.

This heterogeneity across schools may affect our results if correlated with the gender composition of different schools. [11] find that gender differences across sub-disciplines in economics can explain a large share of the gender gap in obtaining assistant professor positions

**Table 1. Descriptive statistics by school.**

|  | Promotions | Female | PhD UNIGE | Quality | Obs.. |
|---|---|---|---|---|---|
| University of Geneva | 0.38 | 0.37 | 0.36 | 41.0 | 750 |
| Medicine | 0.42 | 0.29 | 0.43 | 52.5 | 355 |
| Sciences | 0.30 | 0.31 | 0.19 | 49.8 | 142 |
| Psychology&Education | 0.34 | 0.51 | 0.48 | 34.7 | 59 |
| Literary Arts | 0.44 | 0.42 | 0.38 | 12.3 | 55 |
| Economics&Management | 0.45 | 0.41 | 0.22 | 23.1 | 49 |
| Social Sciences | 0.21 | 0.52 | 0.21 | 16.2 | 44 |
| Law | 0.44 | 0.65 | 0.57 | 7.5 | 23 |
| Translation | 0.25 | 0.75 | 0.44 | 12.1 | 16 |
| Theology | 0.43 | 0.57 | 0.29 | 8.4 | 7 |

Note: The descriptive statistics provide the mean value for the period 2004-2023 in a sample of 750 assistant and associate professors at the University of Geneva. Promotions, in the first column, is the share of faculty that obtained a promotion. Female, in the second column, is the share of female faculty. PhD UNIGE, in the third column, is the share of faculty that obtained a PhD from the University of Geneva. In the fifth column, Quality is measured as the ratio of citations to published papers. The number of observations is given in the last column.

within economics. In the case of promotions, the gender gap is persistent after controlling for disciplines. We will address this by using school-fixed effects when estimating the proportional hazard survival models of the probability of promotion. In the case of research quality, we could have normalized the research quality of each researcher by the average research quality in the discipline. Note, however, that because we are taking the natural log of research quality, this would lead to identical estimates obtained using discipline-fixed effects.

Ideally, we would have liked to estimate the gender bias for each discipline, but the low number of women promoted in our sample would leave us with very limited statistical power. Indeed, the average number of women promoted across schools during the period 2004-2023 is 10.2, which would require heroic assumptions to identify a gender bias. Thus, we are left with the identification of the average gender bias across all disciplines.

Finally, we would also have liked to estimate the gender bias for different cohorts to examine whether the gender bias declines with time. Again, we suffer from the low number of women promoted in each cohort. The average cohort has 4.6 female researchers being promoted, making this task impossible. Note that when we interact a time trend with the gender dummy, we find the interaction to be positive, suggesting that the gender bias declines with time, but the point estimate is not statistically different from zero.

## Results and discussion

We first examine how the promotion hazard evolves over time for female and male faculty using Kaplan-Meier survival plots. Fig 1 provides these plots by gender for the full sample on the left panel and for the subsample of assistant professors on the right panel.

Both panels show a systematically lower probability of promotion for female faculty throughout the 19-year maximum period in the sample (2004-2023), as the solid line for female professors is always below the dotted line for male professors. The gender gap for assistant professors on the right panel seems higher than in the full sample on the left panel, particularly in the early years. After 5 years, around 40 percent of male assistant professors were promoted, but no female assistant professor was promoted. In the full sample, the difference is

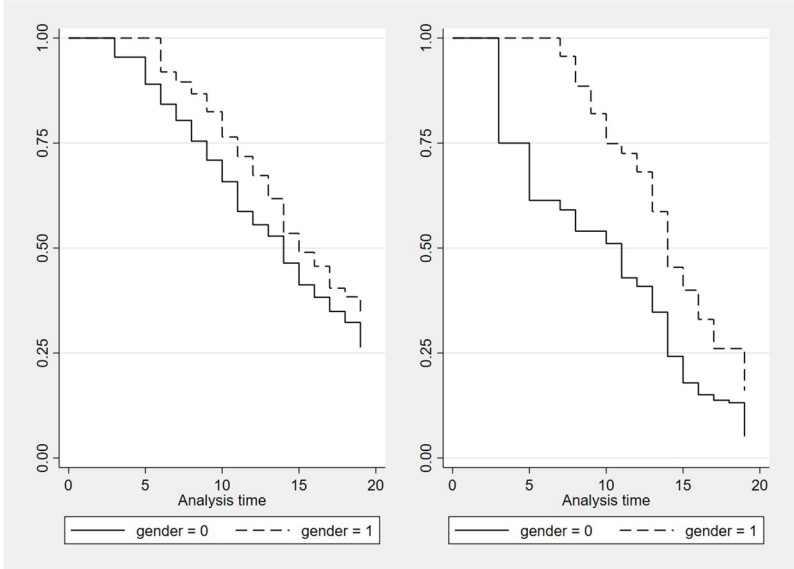

**Fig 1.**

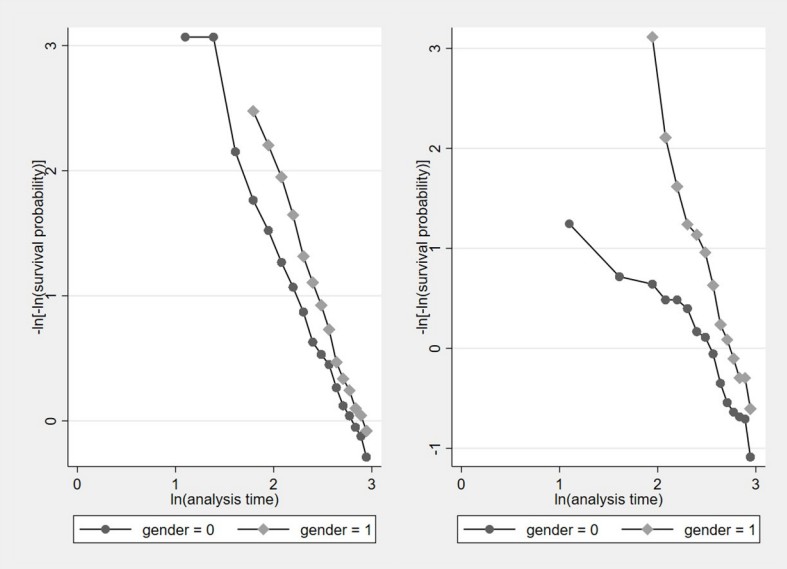

**Fig 2.**

smaller, with still no female professors being promoted after 5 years, but only 13 percent of male professors are promoted after 5 years.

Fig 2 shows the test of proportional hazard for the full sample on the left panel and for the subsample of assistant professors on the right panel. The curves for female and male faculty do not cross, suggesting that the proportional hazard assumption holds.

While the differences between male and female promotion hazards are larger for assistant professors than in the full sample, they are not statistically significant in either sample. The null assumption that the survivor functions are equal across gender is never rejected regardless of the type of test (log rank rest, Wilcoxon–Breslow–Gehan test, or a Peto–Peto–Prentice test as provided by the *st test* command in *Stata*.), or whether we use the full sample or the subsample of assistant professors. This may be partly due to heterogeneity across disciplines or schools.

A problem with the Kaplan-Meier survival plots in Fig 1 is that they do not control for differences across disciplines. We also want to control for other determinants of promotion, such as the quality of the research output and whether the researcher has a PhD from the University of Geneva, which may also be correlated with gender and, therefore, bias our results. There are of course other criteria considered in promotion decisions, such as the quality of teaching, community service and management. These are, however, difficult to observe, and we will (courageously) assume that they are uncorrelated with gender (and some may argue also uncorrelated with promotion). To be able to control for these characteristics, we adopt the semiparametric Cox survival model and the parametric Weibull hazard model.

Table 2 provides the estimates of both the Cox and Weibull models with and without controls in the full sample and the subsample of assistant professors.

The AIC suggests that the parametric Weibull model provides estimates that are closer to the unobserved true model, as the AIC of the Weibull model are several orders of magnitude smaller than the ones provided by the Cox models, regardless of whether we use controls or different samples. The likelihood ratio statistics used to decide whether the model with control variables is a better model suggest that both in the full sample and the assistant professor

**Table 2. Determinants of promotion hazard.**

| | Full sample | | | | Assistant professors | | | |
|---|---|---|---|---|---|---|---|---|
| | Cox | Weibull | Cox | Weibull | Cox | Weibull | Cox | Weibull |
| Gender | 0.87★★ | 0.87★★ | 0.91★ | 0.89★★ | 0.78★★ | 0.79★★ | 0.72★★ | 0.69★★★ |
| | (0.05) | (0.06) | (0.05) | (0.05) | (0.08) | (0.07) | (0.06) | (0.07) |
| PhD | | | 1.44★★ | 1.45★★ | | | 1.51★★ | 1.56★★★ |
| | | | (0.25) | (0.26) | | | (0.21) | (0.25) |
| Quality | | | 1.44★★★ | 1.47★★★ | | | 1.44★★★ | 1.51★★★ |
| | | | (0.12) | (0.14) | | | (0.15) | (0.16) |
| School f.e. | No | No | Yes | Yes | No | No | Yes | Yes |
| LogLikelihood | -1477 | -277 | -1443 | -251 | -397 | -30 | -378 | -15 |
| AIC | 2957 | 560 | 2909 | 527 | 797 | 67 | 781 | 57 |
| Observations | 674 | 674 | 668 | 668 | 226 | 226 | 221 | 221 |

Note: The first four columns used the entire sample, and the last four columns focused on tenure decisions for assistant professors. Gender takes the value 1 if the researcher is female and zero otherwise. PhD takes the value 1 if the researcher obtained his PhD at the University of Geneva and zero otherwise. Quality is the measure of researcher quality as the ratio of citations and the number of papers. The odd columns provide the results of the estimation of a Cox proportional hazards model through maximum likelihood. The even columns provide the results of the maximum likelihood estimation of a parametric survival-time Weibull model. All coefficients are reported as hazard ratios. Numbers in parentheses are standard errors of the hazard ratio estimates clustered by school.

★ stands for 10 percent statistical significance,

★★ stands for 5 percent statistical significance, and

★★★ stands for 1 percent statistical significance.

AIC is the Akaike Information Criteria, where $AIC = -2LogLikelihood + 2(c + p + 1)$, with $c$ being the number of covariates, and $p$ is the number of parameters necessary to estimate each model (0 for Cox and 1 for Weibull regressions).

subsample control variables should be introduced (even if the differences in the estimates of the gender coefficient are not statistically different from each other). The likelihood ratio statistic for the Weibull model equals 52 in the full sample and 20 in the subsample of assistant professors. Both statistics are higher than the 5 percent critical value for the chi-squared distribution with 10 degrees of freedom (PhD, research quality and eight dummies for schools) of 18. Thus, we can concentrate on the results in the fourth column for the full sample and in the last column for the subsample of assistant professors.

All coefficients in the fourth and last columns are statistically different from 1 at the 5 percent level. Because coefficients are expressed as hazard ratios, their statistical significance is tested against 1 and not 0. When the coefficient is below 1 as is the case for the gender dummy, it means that women are less likely to be promoted. The results in the fourth column for the full sample indicate that female faculty is 11 per cent less likely to be promoted. In the case of junior faculty, the last column shows that female faculty are 31 percent less likely to be promoted.

Interestingly, having a PhD from the University of Geneva increases the probability of being promoted by 45 percent in the full sample and 56 percent in the sample of assistant professors. Assuming these differences are cumulative, this implies that female faculty with a PhD from another University than the University of Geneva is 56 percent less likely to be promoted than male faculty with a PhD from the University of Geneva in the full sample. In the subsample of assistant professors the differences are even larger with female faculty with a PhD from another university being 87 percent less likely to be promoted than a male faculty with a PhD from the University of Geneva.

Research quality impacts the promotion hazard positively both in the full sample and the assistant professor sample. The estimates of the hazard coefficients are large. They suggest that

**Table 3. Exploring the mechanisms.**

|  | (1) | (2) | (3) | (4) |
|---|---|---|---|---|
| Gender | 0.89★★ | 1.56 | 0.86★ | 1.52 |
|  | (0.05) | (0.58) | (0.07) | (0.43) |
| PhD | 1.45★★ | 1.46★★ | 1.42 | 1.44★ |
|  | (0.26) | (0.25) | (0.32) | (0.31) |
| Quality | 1.47★★★ | 1.54★★★ | 1.47★★★ | 1.54★★★ |
|  | (0.14) | (0.11) | (0.13) | (0.11) |
| Gender×Quality |  | 0.87★ |  | 0.88★ |
|  |  | (0.07) |  | (0.07) |
| PhD×Quality |  |  | 1.07 | 1.04 |
|  |  |  | (0.45) | (0.22) |
| School f.e. | Yes | Yes | Yes | Yes |
| LogLikelihood | -251 | -250 | -251 | -250 |
| Observations | 668 | 668 | 668 | 668 |

Note: Gender takes the value 1 if the researcher is female and zero otherwise. PhD takes the value 1 if the researcher obtained his PhD at the University of Geneva and zero otherwise. Quality is the measure of researcher quality as the ratio of citations and the number of papers. The first column reproduced column (4) in Table 2 (our preferred specification). The second column introduces the interaction of the gender dummy with research quality. The third column introduces the interaction of the gender dummy with the dummy, indicating that the PhD was obtained at the University of Geneva. The fourth column introduces both interactions. All columns use a maximum likelihood estimation of a parametric survival-time Weibull model. All coefficients are reported as hazard ratios. Numbers in parentheses are standard errors of the hazard ratio estimates clustered by school.

★ stands for 10 percent statistical significance,

★★ stands for 5 percent statistical significance, and

★★★ stands for 1 percent statistical significance.

an increase of 1 in our measure of research quality increases the probability of promotion by 47 percent in the full sample. Note that because we measure research quality with an inverse hyperbolic sine transformation of citations($n$) per paper ($p$), a 1 unit increase in research quality implies a $\sqrt{(n/p)^2 + 1}$ increase in citations per paper [12]. This implies that the relationship between citations per paper and the probability of promotion is concave, i.e., the impact of citations per paper declines as citations per paper increases, and at the mean of the sample, a 1 unit increase in research quality is equivalent to an increase of 41 citations per paper.

Perhaps surprisingly, the introduction of control variables does not statistically affect the estimates of the coefficient on gender. This is at odds with results in the literature on the gender wage gap [13], but note that most of the characteristics that this literature control for, such as occupation, education, region, union-coverage, industry etc, are already controlled for in our sample of faculty at the University of Geneva. In the full sample, the coefficient on gender slightly increases after introducing control variables, whereas in the subsample of assistant professors, it decreases, but both differences are not statistically significant. This result echoes [5, 7], who also found that the gender promotion gap persists after controlling for differences in researchers' output. This is not necessarily because the control variables are uncorrelated with gender but can be explained by different correlations across control variables. For example, the introduction of research quality reduces the gender bias (although not statistically so), whereas the introduction of school fixed effects increases the bias (although, again, the coefficients are not statistically different).

Finally, Table 3 explores different mechanisms that could explain the gender gap in promotions. We do this by interacting the gender dummy with research quality and whether the

researcher obtained a PhD from the University of Geneva to see whether gender differences in the returns to research quality or a PhD from the University of Geneva can be the mechanisms driving the gender gap. A potential explanation for the gender gap in the case of junior faculty could be due to a gender difference in the share of tenure and non-tenure-track positions at the assistant professor level, given that the former should facilitate promotion. However, this is not a mechanism through which the gender gap in promotions works: if anything the percentage of women in tenure track positions is higher than that of women in non-tenure track positions at the University of Geneva. In column 4 of Table 3, where both interactions are introduced into our preferred specification in the full sample (column 4 of Table 2 reproduced in column 1 of Table 3), we see that the odds ratio coefficient on the interaction of gender with research quality is smaller than 1 and statistically significant.

The coefficient on the interaction of gender and research quality being smaller than 1 in column 4 of Table 3 suggests that a similar increase in research quality leads to a smaller increase in the women's likelihood of promotion. This echoes the results in [4], who find that the same positive reference letter attributes lead to a lower increase in the probability of being hired for an academic job. Interestingly, the coefficient on the gender dummy becomes statistically insignificant in columns (2) and (4), suggesting that the gender gap is explained by gender differences in how research quality affects the likelihood of promotion. The interaction of the gender dummy with the dummy indicating whether the PhD was obtained at the University of Geneva is statistically insignificant in columns (3) and (4), and as can be seen from column (3), it does not seem to affect the gender gap as the coefficient on the gender dummy is still statistically smaller than 1 and not statistically different from the coefficient on the gender dummy in the first column of Table 3.

Importantly, while the coefficient on the gender dummy becomes statistically insignificant when we introduce the variable interacting gender and research quality in (2) and (4), it becomes larger than 1, signaling that women are more likely to be promoted, everything else equal. This is consistent with the finding of positive discrimination in academy membership in [3] and in obtaining junior academic jobs by [4].

## Conclusion

We examined gender differences in faculty promotion at the University of Geneva using proportional hazard survival models and found that female faculty were 11 percent less likely to get promoted than male faculty after controlling for research quality, whether the researcher has a PhD from the University of Geneva and discipline. In the case of promotions from assistant to associate professors, the gender gap is almost three times larger.

A rationale in the existing literature for the gender gap in academic outcomes is that there is bias in the recognition of the work by female researchers (see [4, 6, 8]). In this paper, we find support for this potential mechanism as an equal increase in research quality leads to a smaller increase in the probability of promotion for women.

The literature offers other potential explanations for the gender gap in faculty promotions. For example, those making the promotion decisions may be consciously or unconsciously reluctant to have women or individuals who are not similar to them in powerful positions (see [14]). If this were the case, we would expect lower discrimination in promotion decisions for junior faculty, as this leads to positions with relatively weak power. The opposite is observed.

Another potential explanation is that female faculty members tend to spend time on tasks that are less likely to lead to promotion, such as participating in commissions or writing reports [15]. While it is straightforward that time spent in these activities leads to fewer

citations and publications, it is less obvious that this would be correlated with the number of citations per paper, which does not exhibit a gender gap.

The evidence in this paper suggests that the driving force behind the gender gap in faculty promotion is a gender bias in the way research quality is assessed. The use of more objective metrics of research quality which do not suffer from a gender gap is an avenue to be explored.

This recommendation may seem to go against women's preference for not using bibliometrics criteria when considering promotion, as identified by [16]. However, their study asked questions regarding citations and the number of publications. Our findings suggest that rather than the number of papers or citations that suffer from well-known gender gaps identified in the existing literature, the number of citations per paper may be a more appropriate measure of research quality. This echoes the recent recommendation in [17] to adopt "mathematical techniques like normalization and standardization to address biases and ensure fair comparisons in bibliometric data."

Our findings also suggest that controlling for the gender differential in the assessment of research quality, women are more likely to be promoted than men (even though this result is not statistically significant). There are two important points that need to be made here. First, the observed positive gender discrimination is not sufficiently large to compensate for the gender differential in the assessment of research quality, as unconditionally, we observe a gender gap. Second, as [18] argues, two wrongs do not necessarily make a right. The first best is to address the gender differential in assessing research quality.

Finally, a potential explanation for the stronger gender discrimination against younger researchers could be fertility decisions. This is not something we could address as we do not have data on the number of children or their birth dates (controlling for the age of the researchers as a proxy for fertility does not affect results, and it is not statistically significant). Even though recent literature based on qualitative data suggests that work-life decisions do not explain the gender bias in academia and that the culprit is the work environment (e.g., [19]), our future research will try to disentangle these forces quantitatively. Indeed, as suggested by [20], data-driven approaches are important to try to understand and offer solutions to gender biases in academia.

## Acknowledgments

We are grateful to Tony Berrada, Eva Cantoni, Klea Faniko, Giovanni Ferro-Luzzi, Yves Flückiger, Delphine Gardey, Patrick Gaule, Virginie Hamel, Juliette Labarthe, Catherine van der List, Laurent Ott, Tiburce Pégatoquet, Michele Pellizzari, Kerstin Preuschoff, José Ramirez, Jun Robert-Nicoud, Steven Wooding, the members of the University of Geneva's Equality Delegation and Commission, an anonymous referee and the Editor, Paolo Ghinetti, for helpful discussions, comments and suggestions, as well as to Paweł Dębski for his help scrapping the citation data from the Publish or Perish dataset.

## Author Contributions

**Conceptualization:** Agata Czech, Marcelo Olarreaga, Olivia Peila.

**Data curation:** Olivia Peila.

**Formal analysis:** Agata Czech.

**Methodology:** Marcelo Olarreaga.

**Project administration:** Marcelo Olarreaga.

**Supervision:** Marcelo Olarreaga.

**Validation:** Marcelo Olarreaga.

**Writing – original draft:** Agata Czech, Marcelo Olarreaga.

**Writing – review & editing:** Marcelo Olarreaga, Olivia Peila.

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
