## [Decision Letter · Decision Letter 0]

22 Aug 2024

PONE-D-24-20495Gender Gap in Faculty PromotionPLOS ONE

Dear Dr. Olarreaga,

Thank you for submitting your manuscript to PLOS ONE. After careful consideration, we feel that it has merit but does not fully meet PLOS ONE’s publication criteria as it currently stands. Therefore, we invite you to submit a revised version of the manuscript that addresses the points raised during the review process. In particular, the reviewer has concerns about regarding some aspects of the empirical strategy that should better address and clarify. After my own reading of the paper, I agree with the referee on the main comments. First, a time trend in the gender gap in faculty promotion is not controlled for. Second, research quality is also time invariant and measured just before entering UNIGE, which is an issue given that the research quality can largely vary over time and differentially by gender. Third, the empirical strategy does not control for age of the researcher and therefore the larger effects found among junior faculty can be related to fertility decisions often occurring at an early stage of the career of a researcher.

We look forward to receiving your revised manuscript.

Kind regards,

Paolo Ghinetti, PhD

Academic Editor

PLOS ONE

Journal Requirements:

Reviewers' comments:

Reviewer's Responses to Questions

**Comments to the Author**

1. Is the manuscript technically sound, and do the data support the conclusions?

Reviewer #1: Yes

2. Has the statistical analysis been performed appropriately and rigorously? 

Reviewer #1: Yes

3. Have the authors made all data underlying the findings in their manuscript fully available?

Reviewer #1: Yes

4. Is the manuscript presented in an intelligible fashion and written in standard English?

Reviewer #1: Yes

5. Review Comments to the Author

Reviewer #1: Comments on the paper “Gender Gap in Faculty Promotion”

This paper uses data from the University of Geneva from 2004 to 2023 to study the gender gap in faculty promotion using a parametric hazard survival model and addressing the measurement error in assessing research quality. The paper finds that – after controlling for research quality, discipline and whether or not the PhD was obtained at the University of Geneva, female faculty members are 11 percent less likely to be promoted on average, with larger effect among junior faculty promoted from assistant to associate level. Heterogeneity by research quality reveals that an increase in research quality among female faculty members has lower return in terms of probability to be promoted.

The paper is interesting and uses an innovative way to address the biases in measuring research quality. However, I have concerns regarding some aspects of the empirical strategy that the authors should better address and clarify. First, despite having data spanning from 2004 to 2023 the authors seem to use the data cross sectionally without exploiting or controlling for a time trend in the gender gap in faculty promotion. This is especially important given the gender equal policies implemented in the least few decades within universities. Second, and relatedly, research quality is also time invariant and measured just before entering UNIGE. This is also a potential issue given that the research quality can largely vary over time and differentially by gender. Third, the empirical strategy does not control for age of the researcher and therefore the larger effects found among junior faculty can be related to fertility decisions often occurring at an early stage of the career of a researcher.

Below, I provide some comments, which I hope the authors will find useful.

Major comments:

1. The paper uses data from 2004 to 2023. However, the empirical strategy does not control for or exploit a time trend in the gender gap in faculty promotion. This is especially relevant in this context given the gender equal policies implemented in the last few decades in universities. The authors should show whether there is a trend over time in the gender gap in faculty promotion and how this has changed in the 19 years of data used by the authors.

2. Data to construct research quality is measured at the time of entry into UNIGE, therefore it is a time-invariant measure. This is a potential limitation given that the research quality can largely vary over time and differentially by gender. Faculty members could enter UNIGE at a very early stage in their career when their research output is still relatively small. For example the six years of tenure track are normally very productive in terms of research output, potentially different by gender and this of course can affect promotion. The authors should control for a measure of research quality measured just before applying for tenure. Alternatively the authors should describe the evolution of research quality by gender over time once joined UNIGE.

3. The empirical strategy does not control for age of the researcher and therefore the larger effects found among junior faculty can be related to fertility decisions often occurring at an early stage of the career of a researcher. The authors should control for age of the researcher as well as for fertility decisions. This would be useful to explain some of the channels explaining the gender gap in promotion especially among junior faculty.

4. The empirical strategy takes into account that there might be effects that are school correlated with the gender composition of the school and the promotion across gender. However, these effects that are school specific are very likely to change over time over 19 years of observations. This is another aspect that emphasize that the time effects should be taken into account in the strategy – see point 1 above.

Minor Comments:

• The paper uses an innovative and convincing way to address the biases in measuring research quality. For this reason I would suggest to emphasize this aspect more in the abstract as well as in the intro by anticipating the analysis provided in support of the unbiasedness of their measure.

• Table 1 shows that 38 percent of faculty obtained a promotion in the period 2004-2023. This number seems quite low, considering the 19 years of observations. How does this compare to other universities in Switzerland and other countries?

• To support the findings in Table 3 (end of page 12) the authors should show how the gender gap in hiring has changed over the same period of time.

6. PLOS authors have the option to publish the peer review history of their article (what does this mean?). If published, this will include your full peer review and any attached files.

Reviewer #1: No

---

## [Author Response · Author response to Decision Letter 0]

7 Oct 2024

View the attached cover letter and the file Response to reviewers with detailed responses to the reviewer and editor's comments

---

## [Decision Letter · Decision Letter 1]

23 Oct 2024

Gender Gap in Faculty Promotion

PONE-D-24-20495R1

Dear Dr. Olarreaga,

We’re pleased to inform you that your manuscript has been judged scientifically suitable for publication and will be formally accepted for publication once it meets all outstanding technical requirements.

Kind regards,

Paolo Ghinetti, PhD

Academic Editor

PLOS ONE

Additional Editor Comments (optional):

Reviewers' comments:

Reviewer's Responses to Questions

**Comments to the Author**

1. If the authors have adequately addressed your comments raised in a previous round of review and you feel that this manuscript is now acceptable for publication, you may indicate that here to bypass the “Comments to the Author” section, enter your conflict of interest statement in the “Confidential to Editor” section, and submit your "Accept" recommendation.

Reviewer #1: All comments have been addressed

2. Is the manuscript technically sound, and do the data support the conclusions?

Reviewer #1: Yes

3. Has the statistical analysis been performed appropriately and rigorously? 

Reviewer #1: Yes

4. Have the authors made all data underlying the findings in their manuscript fully available?

Reviewer #1: Yes

5. Is the manuscript presented in an intelligible fashion and written in standard English?

Reviewer #1: Yes

6. Review Comments to the Author

Reviewer #1: The authors have thoroughly tried to address the concerns described in the report. Despite the data limitations, the authors have included a clear discussion in the manuscript of such potential limitations and this makes the paper more complete. I have no remaining comments to be addressed.

7. PLOS authors have the option to publish the peer review history of their article (what does this mean?). If published, this will include your full peer review and any attached files.

Reviewer #1: No

---

## [Editor Report · Acceptance letter]

29 Oct 2024

PONE-D-24-20495R1 

PLOS ONE

Dear Dr. Olarreaga, 

I'm pleased to inform you that your manuscript has been deemed suitable for publication in PLOS ONE. Congratulations! Your manuscript is now being handed over to our production team.

Kind regards, 

on behalf of

Professor Paolo Ghinetti 

Academic Editor

PLOS ONE